# Effects of Hearing Impairment, Quality of Life and Pain on Depressive Symptoms in Elderly People: A Cross-Sectional Study

**DOI:** 10.3390/ijerph182212265

**Published:** 2021-11-22

**Authors:** Weiyi Sun, Teruyuki Matsuoka, Ayu Imai, Nozomu Oya, Jin Narumoto

**Affiliations:** Department of Psychiatry, Graduate School of Medical Science, Kyoto Prefectural University of Medicine, Kyoto 602-8566, Japan; sunweiyi@koto.kpu-m.ac.jp (W.S.); imaiayu@koto.kpu-m.ac.jp (A.I.); n-oya@koto.kpu-m.ac.jp (N.O.); jnaru@koto.kpu-m.ac.jp (J.N.)

**Keywords:** visual impairment, hearing impairment, depression, aging, quality of life, pain, social contact

## Abstract

Reduction of the incidence of depression and improvement of quality of life (QOL) of elderly people have become important subjects. Various factors are related to depressive symptoms in elderly people, and sensory impairment is a key to health, QOL, and depressive symptoms. In this cross-sectional study, a visual acuity test and audiometry were used to examine the relationships of visual and hearing impairment and other factors with depressive symptoms in elderly people. A group of 490 community-dwelling people aged over 65 years old underwent vision and hearing tests, the Center for Epidemiologic Studies Depression Scale (CES-D), Mini-Mental State Examination (MMSE), and questionnaires on social isolation, QOL, and physical condition. Logistic regression analysis was used to examine factors affecting CES-D. Multivariate logistic regression analysis indicated that hearing impairment, pain score, and satisfaction with human relationships and activities of daily living (ADL) were independent predictors of CES-D scores. Satisfaction with human relationships and ADL can reduce depressive symptoms in elderly people. Hearing impairment, pain, and social contact are also important. Therefore, improvement of social networks, interpersonal relationships, ADL, and hearing impairment may be effective in improving these symptoms in elderly people.

## 1. Introduction

Depression is an important factor affecting the mental health of elderly people. According to World Health Organization statistics, 322 million people worldwide suffered from depressive disorders, including major depressive disorder and dysthymia, in 2015, accounting for 4.4% of the global population [1]. Between 2005 and 2015, the number of people with depressive disorders increased by 18.4% [1]. This represents not only population growth, but also a rise in the prevalence of depressive disorders in people of all ages. Depressive disorders increase with age, with 7.5% of older women aged 55 to 75 years and 5.5% of older men in the general population becoming depressed. Therefore, depressive disorders are more common in elderly people than in adults and teenagers [1]. In Japan, the situation is similar: the prevalence of depressive disorders was 4.2% in 2015 [1] and mood disorders, including depressive disorders and bipolar disorders, in people aged 65 or older was the highest among all generations in 2017 [2].

In older adults, mild depressive symptoms are often undetected, and one of four individuals with minor depressive disorder will experience major depressive disorder within two years [3]. A previous review suggested that depression is also a risk factor for dementia [4]. Subthreshold depression, which does not meet the criteria for depressive disorder and is the most common symptom of mild behavioral impairment, also increases the risk of dementia [5]. In addition, with increasing aging in Japan, improving the mental health of elderly people has become a top priority [2]. Thus, reduction of depressive symptoms and improvement of quality of life (QOL) of elderly people has become an important area of study.

Psychosocial factors are related to depressive symptoms in elderly people, including social isolation [6], loneliness [7,8], and QOL [9]. Psychosocial factors such as economic hardship, injury, loneliness, migration, caregiving, and bereavement might lead to physiological changes, including abnormalities in the limbic structures and frontostriatal circuit, and increase susceptibility to depression or cause depression in elderly people who are already fragile [3]. In addition, physical conditions such as cardiovascular disease [10], chronic pain [11], and cancer [12] can also lead to the appearance of depressive symptoms in older people by affecting both physiological changes and psychosocial factors [3].

Sensory loss is common in elderly people, with a recent American study finding at least one of visual, hearing, olfactory, taste and tactus impairment in 94% of elderly people; with two of these impairments in 38% and three to five in 28% [13]. Hearing and vision impairment are common among elderly people [14] and are related to depressive symptoms [15,16,17,18,19,20], anxiety [18], emotional vitality [21], life satisfaction [17,19], loneliness [19,22], social isolation [17,19,22,23], self-esteem [19], autonomy [19], and cognitive impairment [24]. Although hearing and vision impairment are associated with depressive symptoms, this relationship might be qualified by various factors. There is evidence that older adults with both hearing and visual impairment have worse depressive symptoms, life satisfaction, positive and negative emotions, loneliness, social isolation, self-esteem, and autonomy among psychosocial factors [19]. Elderly people with impaired vision are more likely to report feeling discriminated against in public, and those who said they had been suffered discrimination had more depression symptoms and isolation, as well as poorer QOL and life satisfaction [17]. Elderly females with hearing loss also have more depressive symptoms than people without hearing loss [15].

Vision or hearing impairment has been widely shown to promote depression in elderly people [15,16,17,18,19,20], but the criteria for impairment in these studies have not been uniform. A few studies have assessed visual or hearing loss using a visual acuity test [13] or audiometry [15,21], while many have assessed these impairments by interview or questionnaire [14,16,17,18,19,22,23,24]. In our previous study [8], visual and hearing impairment assessed retrospectively were found not to be related to depressive symptoms in people with dementia, and only loneliness was involved in these symptoms. However, this may be due to inaccurate evaluation of visual and hearing impairment. Moreover, the causes of depressive symptoms in elderly people, such as sensory loss, cognitive impairment, social isolation, QOL, and physical condition, may be intertwined. Thus, in the present study, we used both a visual acuity test and audiometry to examine the relationships of visual and hearing impairment, cognitive impairment, social isolation, QOL, and physical condition with depressive symptoms in elderly people.

## 2. Materials and Methods

### 2.1. Participants

The Department of Epidemiology for Longevity and Regional Health, Kyoto Prefectural University of Medicine, has undertaken a cohort study in Kyotango city to examine longevity in people aged ≥ 65 years old. In this cohort, the participants are community-dwelling and can visit a hospital for various examinations. Therefore, the participants are relatively healthy people. Data at baseline for 490 people in this cohort were used in the present study. All subjects underwent a visual acuity test and audiometry; the Center for Epidemiologic Studies Depression Scale (CES-D) [25]; Mini-Mental State Examination (MMSE) [26]; and questionnaires on social isolation, QOL, and physical condition on the same day. The cohort study was approved by the ethics committee of Kyoto Prefectural University of Medicine. All participants gave informed consent to participation in the study.

### 2.2. Assessment

At the initial visit, age, gender, and years of education were recorded. Education level was defined as a rank classification variable (1: elementary school, 2: junior high school, 3: high school, 4: junior college/technical college, 5: college).

Depressive symptoms were assessed using the CES-D, a 20-item self-administered questionnaire. On a 4-point Likert scale ranging from 0 (none) to 3 (five days or more), participants are asked to rate the frequency with which each item occurs every week. Higher scores indicate more severe depressive symptoms [25].

Vision and hearing examinations were performed for all subjects. Visual acuity was measured using the Landolt ring scale and the result is presented as a decimal. Visual impairment was defined as <0.3 in the better eye. Audiometry was performed in the low frequency band of 1000 Hz and high frequency band of 4000 Hz, and the decibel (dB) was used as a unit of measurement for display. Hearing impairment was defined as >40 dB at 1000 Hz or 4000 Hz in the better ear. Both visual and hearing scores were converted into dichotomous variables (0: absence, 1: presence).

Cognitive impairment was assessed using the MMSE, which assesses orientation, immediate recall, calculation/attention, delayed recall, naming, repetition, three-stage command, reading, writing, and constructional praxis. The total score is 30, and a lower score indicates severer cognitive impairment [26].

Social isolation was evaluated using living status, marital status, and the number of people the subject met or talked to in a month. Living status was defined as living alone or with family; marital status as married or unmarried (including single, divorced, or widowed). Living alone and marital status were converted into dichotomous variables (0: no, 1: yes).

QOL was examined in a questionnaire, in which satisfaction with human relationships (0: dissatisfied, 1: slightly dissatisfied, 2: slightly satisfied, 3: satisfied) and with activities of daily living (ADL) (0: dissatisfied, 1: slightly dissatisfied, 2: slightly satisfied, 3: satisfied) were evaluated.

Physical condition was assessed with body pain and treatment for cardiovascular disease or cancer. Body pain for the past month was scored as 0 (no pain), 1 (slight pain), 2 (mild pain), 3 (moderate pain), 4 (severe pain), and 5 (extreme pain). Treatment for cardiovascular disease or cancer was assessed at the interview as a dichotomous variable (0: absence, 1: presence).

### 2.3. Statistics

Spearman correlation analysis was used to examine correlations of factors with hearing or visual impairment. Variables affecting CES-D scores were identified using univariate and multivariate logistic regression analysis. A forced entry method was used in multivariate analysis. Dichotomous variables for CES-D scores using a cut-off score of 16 [25] were used as dependent variables, and age, gender, education level, living alone or not, marital status, presence of hearing impairment, presence of visual impairment, number of people the subject met or talked to in a month, satisfaction with human relationships, satisfaction with ADL, MMSE score, pain score, presence of cardiovascular disease, and presence of cancer were included as independent variables. SPSS 26 (IBM Corp., Armonk, NY, USA) was used to analyze the data, and *p* < 0.05 was considered significant in all tests.

## 3. Results

### 3.1. Characteristics of Subjects

The characteristics of the subjects are shown in Table 1. The average age was 73.3 years, and more subjects were female. Cognitive impairment and depressive symptoms were mild. About 16% of the subjects were visually impaired, 34% were hearing impaired, 23% were married, 13% were living alone, 85% were somewhat or completely content with their interpersonal ties, and 70% felt that they could cope with ADL or care for themselves completely. The subjects conversed with an average of 12.5 ± 9.0 people in a month, including family members, relatives, and friends. Some degree of pain was present in 73% of the subjects. Eighteen subjects were under treatment for cardiovascular disease, and 31 were receiving treatment for cancer.

### 3.2. Spearman Correlation Analysis

In Spearman correlation analysis, age (ρ = −0.140, *p* = 0.002) was the only factor related to visual impairment. In contrast, age (ρ = 0.350, *p* < 0.001), education score (ρ = −0.173, *p* < 0.001), CES-D score (ρ = 0.118, *p* = 0.009), and gender (ρ = −0.191, *p* < 0.001) were related to hearing impairment. Visual or hearing impairment was not significantly correlated with cognitive impairment, social isolation, QOL, or physical condition.

### 3.3. Logistic Regression Analysis

The results of univariate analysis are shown in Table 2. In univariate analysis, hearing impairment, pain score, satisfaction with human relationships, satisfaction with ADL, and the number of people met or talked to in a month had an impact on depressive symptoms. Cognitive impairment was not related to CES-D score. In multivariate analysis, hearing impairment, pain score, satisfaction with human relationships, and satisfaction with ADL were identified as independent variables related to depressive symptoms (Table 3).

## 4. Discussion

The results of the study showed that depressive symptoms in elderly people are related to satisfaction with human relationships, satisfaction with ADL, hearing impairment, and pain. Thus, hearing impairment may have an important influence on depressive symptoms, in addition to subjective satisfaction of elderly individuals with their interpersonal connections and ADL.

Hearing impairment worsened with age in our subjects, but visual impairment did not do so. Hearing impairments were more common in male subjects, in subjects with a lower education background, and in those with greater depressive symptoms, consistent with previous studies [15,20,27]. Previously reported risk factors for visual impairment include female, older, and low educational level [28], but these factors were not identified in the present study. This may be attributable to the sample size of 490 participants in this study compared to 4190 subjects in Hu et al. [28]. Additionally, the mean visual acuity of the left and right eyes in our subjects was about 0.5, which is relatively high, and the prevalence of visual impairment of 16% was correspondingly low. This may also have affected the findings for visual impairment.

In our subjects, hearing impairment was a significant factor in development of depressive symptoms, but visual impairment had no effect on these symptoms. More importantly, hearing impairment was not associated with other factors related to depressive symptoms, including pain, satisfaction with human relationships, satisfaction with ADL, and the number of people met or talked to in a month. This indicates that hearing impairment directly affects depressive symptoms in elderly people. Previous studies have shown the influence of hearing impairment [15,18,20] and visual impairment on depressive symptoms in elderly people [16,18]. The discrepancy in our results may be due to the low prevalence of visual impairment, as indicated above. Assessment of visual acuity using the Landolt ring may lead to different findings, since vision measures based on Landolt C charts differ significantly from those in gold standard vision tests for Early Treatment in Diabetic Retinopathy Study [29]. Thus, a further study is needed to examine the relationship between visual impairment and depressive symptoms in elderly people.

Previous studies have shown an association between QOL and depression in elderly people [9]. In people with Alzheimer’s disease, depressive symptoms remain a significant predictor for poor QOL [30]. In Chan et al., QOL was measured using the Quality of Life in Alzheimer’s Disease (QOL-AD) interview, which includes five domains: interpersonal, environmental, functional, physical, and psychosocial status [30]. The current study assessed QOL using a questionnaire with fewer domains, but despite this simpler approach, satisfaction with human relationships and ADL were linked to depressive symptoms in both univariate and multivariate analyses, indicating similar results. Poor QOL may be a result of depressive symptoms, but it is unclear if poor QOL causes depressive symptoms.

Loneliness was not evaluated directly in this study, but satisfaction with human relationships and number of people met or talked to in a month, which might be associated with loneliness, were linked to depressive symptoms. These results are consistent with those in a previous study showing that more depressive symptoms in elderly people in nursing homes were associated with fewer visits from family members, fewer children in a family, a bedridden status, and a lack of additional activity [31]. In the CES-D, some items refer to a lonely feeling [25], although loneliness and depression are regarded as distinct concepts [32]. Loneliness remains as an influential factor for depressive symptoms after controlling for demographic covariates, marital status, and psychosocial factors such as perceived stress, low social support, and hostility [33]. Our previous study identified loneliness as the only predictor for depressive symptoms in patients with dementia, while living alone, visual and hearing impairments, marital status, age, gender, years of education, and cognitive impairment were not predictors [8]. Loneliness can also mediate the relationship between social networks and depressive symptoms: a small social network has an effect on depression only in individuals who feel lonely; and similarly, loneliness is related to having a small social network, particularly among individuals who are depressed [7]. These results show that loneliness has an important effect on depressive symptoms. Thus, expanding social networks and eliminating loneliness is likely to be helpful for reducing depressive symptoms in elderly people.

Pain was identified as another major factor influencing depressive symptoms in elderly people in the present study. Depression and chronic pain are prevalent in older adults, with 13% having both [34]. A comprehensive review of the association between pain and depressive symptoms in elderly people revealed a bidirectional relationship, with each being a risk factor for the other [11]. Therefore, early pain detection and therapy can be highly beneficial for relieving depressive symptoms in elderly people.

One limitation of the study is that the subjects were relatively healthy. In particular, the low prevalences of visual impairment, cardiovascular disease, and cancer may have influenced the results. Additionally, QOL was assessed by questionnaire instead of interview, and the scope was relatively narrow. The association between QOL and depressive symptoms may be more clearly shown with further refinement of the assessment of QOL. Moreover, a causal relationship between depressive symptoms and other factors could not be determined because of the cross-sectional study design. Therefore, further analyses using longitudinal data are needed. However, the assessments of vision and hearing using a visual acuity test and audiometry are a strength of the study, since these tests reduce bias and error caused by self-estimation.

## 5. Conclusions

In conclusion, this study indicates that hearing impairment, QOL, and pain were associated with depressive symptoms in elderly people. QOL may include satisfaction with human relationships, satisfaction with ADL, and the number of people met in a month. Thus, improved hearing through aids, physical pain therapy, assisting subjects in improving interpersonal connections, training to restore everyday functioning, and extending social networks may be effective for improvement of depressive symptoms and mental health in elderly people.

## Figures and Tables

**Table 1 ijerph-18-12265-t001:** Clinical characteristics of the subjects (*n* = 490).

Characteristic	Value
Gender (Male/Female), *n*	210/280
Age, years	73.3 ± 5.6
Education (1: elementary school/2: junior high school/3: high school/4: junior college/technical college/5: college), *n*	2/92/271/57/68
Education score	3.2 ± 0.9
MMSE score	27.2 ± 2.5
CES-D score	10.9 ± 6.8
CES-D (0: <16/1: ≥16), *n*	381/109
Left vision	0.5 ± 0.3
Right vision	0.5 ± 0.4
Visual impairment (+/−), *n*	78/412
Hearing in the left ear at 4000 Hz	40.2 ± 20.2
Hearing in the left ear at 1000 Hz	26.9 ± 17.2
Hearing in the right ear at 4000 Hz	38.2 ± 19.9
Hearing in the right ear at 1000 Hz	27.8 ± 14.9
Hearing impairment (+/−), *n*	168/322
Living alone (+/−), *n*	63/427
Marital status (+/−), *n*	113/377
Satisfaction with human relationships (0: dissatisfied/1: slightly dissatisfied/2: slightly satisfied/3: satisfied), *n*	6/65/316/103
Satisfaction with human relationships score	2.1 ± 0.6
Satisfaction with ADL (0: dissatisfied/1: slightly dissatisfied/2: slightly satisfied/3: satisfied), *n*	14/123/274/79
Satisfaction with ADL score	1.9 ± 0.7
Number of people (including family, relatives, and friends) met or talked to in a month	12.5 ± 9.0
Pain (0: no/1: slight/2: mild/3: moderate/4: severe/5: extreme), *n*	133/78/168/79/30/2
Cardiovascular disease (0: absent/1: present), *n*	472/18
Cancer (0: absent/1: present), *n*	459/31

MMSE, Mini Mental State Examination; CES-D, Center for Epidemiologic Studies Depression Scale; ADL, Activities of Daily Living.

**Table 2 ijerph-18-12265-t002:** Results of univariate analysis.

Variable	OR *	95% CI	*p* Value
Age	1.027	0.990–1.065	0.153
Female	1.462	0.941–2.271	0.091
Education score	0.897	0.707–1.137	0.369
Visual impairment	0.885	0.488–1.607	0.688
Hearing impairment	1.629	1.053–2.519	0.028
Living alone	1.346	0.737–2.458	0.334
Marital status	0.831	0.508–1.360	0.461
Satisfaction with human relationships	0.276	0.187–0.409	<0.001
Satisfaction with ADL	0.248	0.173–0.355	<0.001
Number of people met or talked to in a month	0.965	0.937–0.994	0.019
MMSE score	0.987	0.907–1.075	0.767
Pain score	1.564	1.304–1.876	<0.001
Cardiovascular disease	0.691	0.196–2.430	0.564
Cancer	1.732	0.790–3.797	0.171

* OR: odds ratio for depressive symptoms defined by the Center for Epidemiologic Studies Depression Scale (CES-D). ADL, Activities of Daily Living; MMSE, Mini Mental State Examination.

**Table 3 ijerph-18-12265-t003:** Results of multivariate analysis.

Variable	OR *	95% CI	*p* Value
Age	1.016	0.967–1.068	0.524
Female	1.591	0.912–2.776	0.102
Education score	1.063	0.803–1.408	0.669
Visual impairment	1.001	0.500–2.002	0.999
Hearing impairment	1.754	1.010–3.049	0.046
Living alone	2.388	0.909–6.272	0.077
Marital status	1.644	0.711–3.803	0.245
Satisfaction with human relationships	0.349	0.219–0.557	<0.001
Satisfaction with ADL	0.376	0.248–0.569	<0.001
Number of people met or talked to in a month	0.990	0.959–1.022	0.531
MMSE score	0.985	0.890–1.090	0.770
Pain score	1.376	1.106–1.713	0.004
Cardiovascular disease	0.551	0.129–2.349	0.420
Cancer	1.564	0.609–4.018	0.353

* OR: odds ratio for depressive symptoms defined by the Center for Epidemiologic Studies Depression Scale (CES-D), ADL, Activities of Daily Living; MMSE, Mini Mental State Examination.

## Data Availability

Research data are not shared because the ethics committee of Kyoto Prefectural University of Medicine did not permit it.

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
