# Peer review of "Effects of Hearing Impairment, Quality of Life and Pain on Depressive Symptoms in Elderly People: A Cross-Sectional Study"

_ijerph, 2021, doi:10.3390/ijerph182212265_

Round 1
Reviewer 1 Report
Thank you for providing me this opportunity to review this article.
Authors have done a cross sectional study to show effects of visual impairment and hearing impairment on depressive symptoms in elderly people, but there are more important results that can be gained from this study and this should be added in the title as well so that readers have a better understanding before reading the article.
Overall authors have done a meticulous work but I would like to highlight a few important points that should be addressed.
- Identifying that this is a cross-sectional study in the method section, and giving a timeline of when the tests and questionnaires were conducted, either on the same day or within a few days.
- Given it is a cross sectional study it is important that only correlations can be devised from this data, it should not be misinterpreted as risk factors. Further longitudinal design and data will be needed to comment on this, however current correlations does set a pretense to collect such data for future studies.
- Tables should be clearly defined that it is the OR for Depression defined by CES-D.
- Did any of the participant met MCI or demented criteria by MMSE? if so a sensitivity analysis should be performed by removing those participants from the analysis as it can significantly confound the results.
Reviewer 2 Report
* Title / Abstract:
The study design does not appear either in the title or in the abstract.
* Introduction
The introduction is correct, although I would further expand the existing information on visual / hearing impairment and depressive symptoms.
* Materials and methods
The methodology is correct.
* Results
The results are not very novel. For example, it is already known that having a sensory disability is related to depressive symptoms.
However, it is correct to reaffirm these data with new well-designed studies like yours.
*Discussion and Conclusions
The discussion and conclusion sections are well developed.
*Introduction
The first paragraph sets out the incidence and prevalence of depression worldwide. Depression is reported to be a common problem in older people. Since the study takes place in Japan, it would be interesting to also include prevalence / incidence data in the study population (older people in Japan). In addition, presenting a definition of depression would also be interesting.
In the second paragraph:
Lines 35-36: You speak of "mild depression", "minor depression" and "major depression". Is mild and minor depression synonymous? Can you briefly explain these terms?
In line 37-38 the expression “Subthreshold depression” is also used. Can you specify this term?
Lines 39-41: They also present depression as a risk factor for suicide, but it is not the object of study in this research.
Lines 41-44: They do not have bibliographic references. Which Japanese institution has indicated that mental health is a priority? Who said that the study of depression and quality of life in older people are important?
The third paragraph indicates the factors related to depressive symptoms in older people.
It would be interesting to classify, if possible, the factors into: clinical and socio-economic.
Lines 46-48: Could you indicate what are the “physiological changes” that increase susceptibility or cause depression?
Lines 49-51: Could you briefly explain why these conditions can lead to depressive symptoms?
In the fourth paragraph they expose the relationship between sensory deficits and multiple problems (depression, anxiety, loneliness, etc.).
Lines 56 and 57: The relationship between sensory deficits and depressive symptoms [15-20], I consider that they should be developed in greater depth, since it is the central theme of this research. This would reinforce paragraph 5 where it is stated that this relationship has been amply demonstrated.
Paragraph 5 presents an important limitation (the lack of uniformity in assessing disability in the studies) that justifies the study.
But the objective (paragraph 6) indicates the relationship between (1) visual and hearing impairment, (2) cognitive impairment, (3) social isolation, (4) QOL, and (5) physical condition with depressive symptoms in elderly people.
In general, I think these 5 factors have been named very superficially in the introduction.
*Methodology
In section “2.2. Assessment ”, should better specify how the 5 study factors were assessed:
(1) The "visual and hearing impairment" was measured with vision and hearing tests.
(2) The "cognitive impairment" was assessed using the MMSE. The characteristics of this questionnaire are not detailed.
(3) The “social isolation”: was it evaluated with “the social environment”. Can you specify this?
(4) The “QOL”: How was it measured? With the level of satisfaction with human relationships and with ADLs”? Can you specify this?
(5) The “physical condition”: was it assessed with “body pain” and by “treatment of cardiovascular disease or cancer”?
Consistency of terms is important to be maintained throughout the text.
*Results
The findings of the relationship between: (1) visual and hearing impairment, (2) cognitive impairment, (3) social isolation, (4) QOL, and (5) physical condition with depressive symptoms should be clearly presented. Results are hard to find and some are not presented.
Round 2
Reviewer 1 Report
I would like to congratulate the authors in doing a great work in revising their manuscript. They have addressed most of my concerns, but will still recommend to add the timeline of different tests done in the methodology.